# New Sensing and Radar Absorbing Laminate Combining Structural Damage Detection and Electromagnetic Wave Absorption Properties

**DOI:** 10.3390/s22218470

**Published:** 2022-11-03

**Authors:** Federico Cozzolino, Fabrizio Marra, Marco Fortunato, Irene Bellagamba, Nicola Pesce, Alessio Tamburrano, Maria Sabrina Sarto

**Affiliations:** 1Department of Astronautical, Electrical and Energy Engineering, Sapienza University of Rome, 00184 Rome, Italy; 2Research Center for Nanotechnology Applied to Engineering, Sapienza University of Rome, 00185 Rome, Italy

**Keywords:** graphene-based paint, electromagnetic absorbing material, low observability, EMI suppression, piezoresistive strain sensors, sensor array, structural health monitoring, multifunctional system, aircraft

## Abstract

Within the paradigm of smart mobility, the development of innovative materials aimed at improving resilience against structural failure in lightweight vehicles and electromagnetic interferences (EMI) due to wireless communications in guidance systems is of crucial relevance to improve safety, sustainability, and reliability in both aeronautical and automotive applications. In particular, the integration of intelligent structural health monitoring and electromagnetic (EM) shielding systems with radio frequency absorbing properties into a polymer composite laminate is still a challenge. In this paper, we present an innovative system consisting of a multi-layered thin panel which integrates nanostructured coatings to combine EM disturbance suppression and low-energy impact monitoring ability. Specifically, it is composed of a stack of dielectric and conductive layers constituting the sensing and EM-absorbing laminate (SEAL). The conductive layers are made of a polyurethane paint filled with graphene nanoplatelets (GNPs) at different concentrations to tailor the effective electrical conductivity and the functionality of the material. Basically, the panel includes a piezoresistive grid, obtained by selectively spraying onto mylar a low-conductive paint with 4.5 wt.% of GNPs and an EM-absorbing lossy sheet made of the same polyurethane paint but properly modified with a higher weight fraction (8 wt.%) of graphene. The responses of the grid’s strain sensors were analyzed through quasi-static mechanical bending tests, whereas the absorbing properties were evaluated through free-space and waveguide-based measurement techniques in the X, Ku, K, and Ka bands. The experimental results were also validated by numerical simulations.

## 1. Introduction

Over the past few decades, the aviation and aerospace industry has been increasingly using composite materials that provide lighter weight but still preserve safety, reliability, and longevity standards [1]. However, these materials are exposed to adverse conditions during operating hours. Barely visible impact damages (BVID) on the laminated structures of aircrafts are critical phenomena that need to be analyzed and carefully evaluated [2,3,4].

These impacts can be caused by gravel, hail, or maintenance tools; these are complicated to detect with a simple visual inspection and could lead to either cracks in the structure or fiber breakage to the point of delamination. Given the high quality and stringent safety standards that are required in the aeronautical industry, the components are designed to have a lifetime that exceeds their actual usage, thus incurring more manufacturing costs. Recent studies have focused on the development of novel systems for structural health monitoring (SHM), based on the continuous and distributed monitoring of low-impact occurrences, with the aim of achieving early detection of potential structural damage during aircraft operating time [5,6,7]. This would prevent the late detection of more serious damage, which requires more expensive repairing interventions when possible, or the use of expensive techniques for non-destructive testing and evaluation (NDT&E) [8,9,10]. The use of SHM technologies in the aviation market will have a commercial impact, possibly reducing maintenance costs by up to 40% and also modifying the repairing process, leading to a move from time-based maintenance (TbM) to condition-based maintenance (CbM) [11,12,13].

In general, SHM systems use various types of sensors. Among them, foil strain gauges are widely used [14], although they have limitations in terms of 2D-spatial deformation detection, sensitivity to temperature variation, and fatigue cycles. Another class of sensors are those based on optical-fiber technology, in particular fiber Bragg grating (FBG) [15]. FBG systems provide very accurate responses in cases of highly localized deformations, but they cannot be used for impact detection due to their extreme fragility. FBGs are expensive to repair, and they also require complex and equally costly equipment to allow their use [16].

In the past few decades, the improvement of new materials, and in particular the exploitation of nanotechnologies, has opened up the possibility of producing innovative smart materials and miniaturizing devices in order to provide improved system performance [17,18]. Among the solutions for structural health monitoring using new materials, Sebastian et al. [19] proposed a CNT-based material having piezoresistive properties that can be embedded within the structure to be monitored. This solution could be useful for monitoring large surfaces and reaching places that would otherwise be difficult to access. However, the limitations of this solution lie in the low sensitivity and the high production cost. Graphene-based strain sensors also seem to be of remarkable interest. In this context, Duan et al. [20] described the production of piezoresistive strain sensors by dispersing graphene in a solution of 1-methyl-2-pyrrolidinone and PVDF powder. These thin film sensors can be designed in different configurations, according to the specific application. However, one of the main disadvantages is their low sensitivity, so that a signal amplifier is required in order to distinguish the strain-related signal from noise.

Unlike the various sensors presented above, the development of innovative devices with improved characteristics, such as multifunctional properties, light weight, ease of application, and industrial scalability, is still a challenge.

In addition to SHM systems, in the aerospace industry there is also a considerable interest in the control and suppression of electromagnetic interference (EMI) issues caused by natural events such as lightning, or by the interaction between electrical devices such as antenna and communication systems [21,22,23]. Moreover, the continuous research into innovative solutions in the electromagnetic compatibility (EMC) field has pushed towards the design and development of smart materials and structures able to reduce EMI at radio frequencies. In this context, significant investments come from the defense industry with the aim of studying multifunctional materials able to reduce radar cross section (RCS) and aircraft observability [24] as well as EMI. In the recent literature, Kutluhan et al. [25] investigated the use of nanoparticle-based materials in order to design different solutions to minimize the reflection coefficient and widen the EM absorption bandwidth (BW) at −10 dB. Specifically, they proposed one solution providing a minimum value of the reflection coefficient equal to −28 dB at the frequency of 10 GHz and a second solution with a maximum bandwidth at −10 dB equal to 3 GHz. Taufiq et al. [26] designed a nanostructured material based on Fe_3_O_4_/MWCNT/ZnO for EMC applications in the frequency range between 7 GHz and 13 GHz. The developed sample, having a thickness of approximately 5 mm, was characterized by a central frequency of 10 GHz and an absorption bandwidth (at −10 dB) of 4 GHz. Moreover, Puthucheri et al. [27] proposed a nanostructured nickel-ferrite EM absorber having a total thickness of about 2 mm and a minimum reflection coefficient of −40 dB with a central frequency of 10.3 GHz and a BW at −10 dB of 4 GHz.

In this paper, to meet the need for multifunctional integration of low-impact damage detection and EM energy suppression capability, a novel piezoresistive sensor array based on an aeronautical paint loaded with graphene nanoplatelets (GNPs) [28,29,30] was developed and embedded in a multi-stack laminate, having the structure of a dielectric Salisbury screen (DSS) absorber [31,32]. The DSS basically consists of a lossy dielectric sheet made of the same aeronautical paint filled with GNPs, but at higher concentrations, properly deposited on a mylar substrate.

In other recent work [33,34,35], the authors have tried to solve the combined problems of SHM and EMI using different materials and multifunctional structures. Specifically, in [33], the authors realized networks of SiC fibers embedded in an epoxy resin; Bai et al. [34] constructed a Ni/CNT-decorated melamine sponge; in [35], a robust and porous MXene/polymide aerogel was proposed. Table 1 summarizes the main differences between the solutions investigated in the aforementioned papers and shows some of the characteristics of the new proposed structure that will be analyzed in detail in the following sections.

In particular, the choice of GNPs as nanofiller for the development of the functional paint was due to the fact that the stacked 2D graphene sheets have a high aspect ratio (defined as the ratio of the lateral dimension of the platelet to its thickness) and electrical conductivity, and thus it would allow us to manufacture high-performance strain sensors and radar-absorbing materials [36].

The strip-like nanostructured strain sensors forming the integrated sensing grid were designed, fabricated, and tested under various load conditions to demonstrate their feasibility in performing SHM at low cost and with remarkable ease of application. Indeed, they are simple to manufacture, they can be applied to surfaces with complex geometries, even in hard-to-reach locations, and they also manifest a high degree of sensitivity, capable of detecting very low strains.

The design of the DSS was made through a transmission line theory-based modelling approach, followed by fabrication and experimental characterization. The EM properties of the materials were investigated through rectangular waveguide measurements, in the frequency range between 8 GHz and 26.5 GHz, using different set-ups. The final structure was tested by antenna measurements in order to define the reflection coefficient and the bandwidth at −10 dB in the frequency range 18–40 GHz.

The innovative system described in this work combines sensing and radar-absorbing properties in a unique laminate. In particular, the sensing and EM-absorbing laminate (SEAL) is composed of: (i) an aluminum back panel, which fulfils the structural task and constitutes the conductive reflecting surface of the DSS; (ii) two thin layers of mylar, including the GNP-based piezoresistive strip sensors, rotated 90° with respect to each other, which allow the monitoring of the deformation of the substrate when subjected to quasi-static and cyclic bending tests; (iii) another thin mylar substrate coated with the lossy paint, enabling the radar-absorbing capability.

The EM response of the SEAL, with an overall thickness of only 870 µm (not including the metal back), was characterized by a minimum reflection coefficient of −15 dB at 22 GHz and by a bandwidth at −10 dB of 6.5 GHz, as demonstrated by free-space measurements.

## 2. Materials and Methods

### 2.1. Production of the Nanostructured Paint

The piezoresistive sensors for SHM and the lossy sheet for the radar-absorbing structure were obtained using a commercial aeronautical-grade polyurethane paint properly filled with GNPs. The production process, based on the use of different amounts of fillers, and their optimal dispersion in the dielectric host matrix in order to obtain tailored piezoresistive and EM properties, is sketched in Figure 1a and described in [29].

### 2.2. Strain Sensor Array and Lossy Sheet Realization

The fabrication process of a single piezoresistive strain sensor of the grid is divided into the following steps.

First, two 0.8 cm × 1 cm electrodes are deposited on a 100 µm thick mylar substrate with a silver-based paint. Second, copper wires are attached to each electrode, using a commercially available conductive epoxy glue. Finally, the portions of the electrodes bonded to the wires are masked off and the piezoresistive paint, loaded with 4.5 wt.% of GNPs, is sprayed over the substrate (Figure 1c) using an automated and remote-controlled XY plotter equipped with an airbrush with a 1.2 mm nozzle (Figure 2). The driving-software of the deposition system allows the motion and speed of the plotter, and consequently of the airbrush, to be controlled, tailoring the geometry and thickness of the realized coating. Thus, an active strain-sensitive area of 15.4 cm × 1 cm with a thickness of 70 µm and an overlapping area of 0.4 cm × 1 cm on each electrode is obtained, as shown in the sketch of Figure 3a.

It is notable that the GNP concentration was higher than that used in [28] (4.5 wt.%) because of the increased length of the sensors and the need to contain the electrical resistance of the strips in the order of a few tens of kΩ.

Figure 3b shows a sensor strip on the mylar layer, which was glued to an aluminum beam (24 cm × 2.5 cm) used to perform electromechanical bending tests (see Section 3.2).

The sensors’ grid, devoted to the distributed monitoring of mechanical deformations, was obtained by the spray-deposition of two pairs of strip sensors over two different mylar substrates that were then stacked and positioned at 90° to each other, as shown in Figure 4.

The lossy sheet of the DSS was deposited on another mylar substrate (hereinafter this structure will be referred to as the radar-absorbing structure (RAS)) with the same paint, but filled with a higher weight percentage of GNPs (8 wt.%), as sketched in Figure 1b, to achieve the targeted EM performances.

### 2.3. Assembly of the Multifunctional Sensing and EM-Absorbing Laminate

The sensing and EM-absorbing laminate (SEAL) that includes both the array of piezoresistive sensors and the RAS was assembled once the characterizations of the individual components, with respect to their specific functionality, was completed. As sketched in Figure 5, the two functionalized mylar sheets and the lossy sheet’s substrate constitute together the spacer (having an overall thickness *t_s_*) of the DSS configuration. The various layers were stacked and glued onto an aluminum plate (15 cm × 15 cm × 1 mm), used as a structural and electrically conductive substrate. The final panel is characterized by a total thickness *t_LS_* + *t_s_* + *t_back_* = 1.870 mm (see Section 3.4). Figure 6 presents a picture of the final produced structure.

## 3. Characterizations

### 3.1. Morphological Characterization

The thickness and lateral dimension of the GNPs were studied with atomic force microscopy (AFM) (Dimension Icon AFM, Bruker-Veeco), operating in tapping mode. Figure 7a shows an AFM image of the GNPs with their measured thickness profile (Figure 7b). According to the performed measurements, the produced GNPs had an average thickness of a few tens of nanometers, while the lateral dimensions were on the order of a few microns.

The surface morphology of the GNP coating and the distribution of the filler inside the polyurethane paint was investigated using a field-emission scanning electron microscope (FE-SEM, Auriga, Carl Zeiss, Oberkochen, Germany). The samples were previously coated with a 15 nm thick chromium film by a sputter coater (Quorum Q150T ES) to prevent any surface charges during the characterizations.

Figure 8a–c show, respectively, the surface of the neat coating (consisting of the commercial bicomponent water-based polyurethane paint), the sensor made up with the same paint loaded at 4.5 wt.% of GNPs, and the lossy sheet obtained with the paint loaded at 8 wt.%. All the SEM images showed a good dispersion, a high integration of the nanofiller within the polymer matrix, and an elevated degree of coverage.

Furthermore, in order to demonstrate the good integration of the fillers within the polymer matrix, in Figure 9a,b we show the cross-section and the surface images of the produced samples obtained using the paint loaded at 4.5 wt.% of GNPs. As can be observed, the individual nanoplatelets are easily distinguishable.

### 3.2. Eletromechanical Characterization

The piezoresistive response of the GNP-based paint at 4.5 wt.% was tested with a universal testing machine (Instron 3366) equipped with a load cell of 500 N and a three-point bending test set-up following the ASTM D790 standard [37]. In order to evaluate the electromechanical behavior of the strip-like sensor, quasi-static tests were first performed on samples with the geometry described in Section 2.2. The test was conducted at constant crosshead speed, increasing the load up to a maximum displacement of 2.5 mm and guaranteeing the elastic regime of the aluminum substrate. In particular, the deflection of the device was measured by an extensometer while the electrical resistance variation was monitored using a Keithely 6221 current source and a Keithely 2182A nanovoltmeter. The sensor was evaluated in terms of the resistance variation $\Delta R=R\left(\varepsilon \right)-R_{0}$ [Ω], where R(ε) is the resistance as a function of flexural deformation (ε), and $R_{0}$ represents the resistance measured at rest, with no load applied. Subsequently, the gauge factor GF was calculated as follows:(1)$$GF=\frac{\Delta R}{R_{0}\varepsilon }\;$$

Figure 10 shows the relative percentage variation of the resistance and of the GF as a function of ε. A quasi-linear response was obtained for the former quantity as the strain increased up to 0.05%, with the sensor’s sensitivity reaching the maximum value of 4.

Next, the array with the four sensors on the aluminum plate (see Figure 4) was also tested in order to validate the efficacy of the distributed monitoring. The set-up shown in Figure 11 for the flexural test of the plate was suitably designed with the aid of 3D modelling software and then fabricated to match the dimensions of the sample.

The electrical response of the sensor grid was measured using a DAQ device (USB-6210, National Instruments) properly programmed with a LabView code. Figure 12 shows the circuit diagram of the electrical connections to the data acquisition system: the four piezoresistive sensors are represented by four variable resistors, $R_{S1},R_{S2},\;R_{S3},\;R_{S4}$, powered with a DC voltage ($V_{DC}=5\;V$) through the digital output of the DAQ board. $R_{ref}$ is a reference resistor of known and constant resistance value, and the resistance variation $R_{Si}\left(t\right)$ with respect to time t [s] of the $i^{th}$ sensor when subjected to a mechanical deformation is:(2)$$R_{Si}\left(t\right)=\frac{V_{Si}\left(t\right)}{I\left(t\right)}=\frac{V_{Si}\left(t\right)\;R_{ref}}{V_{DC}-\sum_{j=1}^{4}V_{Sj}\left(t\right)}$$
where $V_{Si}$ is the voltage across the $i^{th}$ sensor measured by the DAQ and I(t) [A] is the current flowing through the series resistors.
(3)$$I\left(t\right)=\frac{V_{DC}-\sum_{j=1}^{4}V_{Sj}\left(t\right)}{R_{ref}}$$

Figure 13a represents the model of the flexural test performed on the plate with the sensor array, identifying the direction of the applied distributed load and the spatial location of the four sensors (S1, S2, S3, S4) of the grid. The curves in Figure 13b–d show the normalized resistance variation of each sensor, measured as a function of time, at 5 mm/min (Figure 13b), 10 mm/min (Figure 13c), and 20 mm/min (Figure 13d) crosshead speed.

It should be noted that, as the sensors laying perpendicular to the load (S2 and S3) are subjected to a greater deformation than those parallel (S1 and S4), the latter manifested a much lower resistance variation during the mechanical tests. Moreover, due to the specific load and structure symmetries, the response of S1 (S2) was almost overlapping with that of S4 (S3). Finally, comparing Figure 13b–d, it can be seen that the resistance peaks of all the sensors were obtained at different moments in time because of the different crosshead speeds.

To validate the experimental tests, and in particular the different responses of the parallel and perpendicular strip sensors, 3D finite element method (FEM) simulations were also performed using COMSOL Multiphysics.

The values of the piezoresistive coefficients of the materials were first extracted as functions of strain from the GF curve (Figure 10) and were then used to simulate the electromechanical FEM model of the plate (Figure 4) with all the sensors. Figure 13a shows a map of the *xx* component of the conductivity tensor of the piezoresistive strips when the plate reached the maximum deflection of 2.5 mm: as expected, the most pronounced color variations concern the S2 and S3 sensors. As a further demonstration, Figure 14 shows that the trend of the simulated sensor resistance responses, obtained with the 3D FEM model, are highly comparable with those of Figure 13b.

Finally, the dynamic and good repeatability responses of the sensors were assessed through the performing of a cyclic test: Figure 15a shows the deflection over time, whereas Figure 15b shows the corresponding relative resistance variations.

### 3.3. Electromagnetic Characterization of the Lossy Sheet

The electromagnetic properties of the paint loaded at 8 wt.% of GNPs, constituting the lossy sheet of the RAS, were investigated through a transmission/reflection-based method [38], i.e., by measuring with a two-port vector network analyzer the scattering parameters of the material specimens inserted in three different sets of rectangular waveguides. Specifically, WR90, WR62, and WR42 waveguide sections, with their proper calibration kits, were used to cover the X (8.2–12.4 GHz), Ku (12.4–18 GHz), and K (18–26.5 GHz) bands, respectively. Subsequently, by following the procedure described in [39,40], the effective complex permittivity $(\varepsilon_{LS}=\varepsilon^{\prime }+j\varepsilon^{\prime \prime }$) of the thin coating vs. the frequency was extracted from the measurements. The results were then used to extend their values up to 40 GHz using the second-order Debye model [41]. Figure 16 shows the real and imaginary part of $\varepsilon_{LS}$ in the measured range 8.2–26.5 GHz, as well as the predicted values at higher frequencies.

In previous work [31,40,42,43], the effective complex permittivity of other types of nanocomposites, including GNPs, have been investigated. The use of a water-based polyurethane paint filled with GNPs, and the extraction of the effective complex permittivity in a wide frequency range, is being addressed for the first time in this paper. Typically, by increasing the filler concentration, higher values of the real part and of the imaginary part (in modulus) of the complex permittivity is obtained. In this case, the maximum possible filler concentration has been adopted to ensure the processability of the nanocomposite material and the successful spray deposition of the GNP-based paint (8 wt.%). This choice has allowed us to obtain a high performance RAS with minimal thicknesses, as described in Section 3.4

### 3.4. Design and Testing of the RAS

Once the values of the complex permittivity of the nano-filled paint were identified, the effects of the values of *t_LS_* and *t_s_* of the DSS were investigated through simulations, performed with the additional aims of limiting the total thickness of the lossy sheet and spacer (*t_LS_* + *t_s_*) to less than one millimeter, and to obtain an absorption peak around the center of the frequency band when illuminating the structure by an EM plane wave with normal incidence. The problem was simplified by assuming the metallic back plate to be a perfect electric conducting (PEC) surface and the spacer, constituted by the lossy sheet’s substrate and the two mylar sheets with sensors, to be a homogeneous lossless dielectric layer of thickness *t_s_* having the dielectric constant $\varepsilon_{s}=3.1$ of mylar. The reflection coefficients in dB of the panel as a function of the frequency and for different layer thicknesses are reported in Figure 17 and calculated using:(4)$$R_{dB}=20\log_{10}\left|\frac{Z_{RAS}-\eta_{0}}{Z_{RAS}+\eta_{0}}\right|$$
where $\eta_{0}$ is the wave impedance of free space and
(5)$$Z_{RAS}=\eta_{0}\frac{{\left(\varepsilon_{s}\right)}^{-\frac{1}{2}}\tanh(\gamma_{s}t_{s})+{\left(\varepsilon_{LS}\right)}^{-\frac{1}{2}}\tanh\left(\gamma_{s}t_{s}\right)}{1+\eta_{0}{\left(\frac{\varepsilon_{LS}}{\varepsilon_{s}}\right)}^{\frac{1}{2}}\tanh\left(\gamma_{s}t_{s}\right)\tanh\left(\gamma_{Ls}t_{Ls}\right)}$$
the input impedance of the *RAS* being
(6)$$\gamma_{s}=j2\pi f\sqrt{\mu_{0}\varepsilon_{0}\varepsilon_{s}},\;\;\;\;\;\;\;\;\;\;\;\;\;\;\;\gamma_{LS}=j2\pi f\sqrt{\mu_{0}\varepsilon_{0}\varepsilon_{LS}}$$
the propagation constant of the spacer and of the lossy sheet, respectively.

As described in Section 2.2, a RAS was produced, with *t_s_* = 750 µm and *t_LS_* = 120 µm, and glued onto an aluminum back. The reflection coefficient of the panel was measured in free space using two rectangular antennas in bistatic configuration and connected to a vector network analyzer (Agilent N5245A PNA-X). The results, shown in Figure 17b, demonstrate the good agreement between the values of the reflection coefficient for the simulated absorbing panel (dashed blue line) and the produced prototype being tested (red line). It can be also observed that the minimum reflection coefficient was about −25 dB at 22 GHz and the bandwidth at −10 dB was around 6 GHz.

Finally, a new RAS with the same lossy sheet was produced and assembled with the sensorized layers to obtain the SEAL of Figure 5 and Figure 6, for which the overall thickness *t_s_* of the three attached mylar-based layers was 750 µm, matching the value of the simplified DSS described above.

## 4. Discussion

The EM response of the SEAL, as well as the effect of the integrated sensor grid on the radar-absorbing performance was subject of the subsequent investigation.

As shown in Figure 18, the final laminate was tested in free space using the same set-up described in Section 3.4. Figure 19 compares the measured reflection coefficient of the DSS (without sensors) of Figure 17b with that of the SEAL, having the same lossy sheet and total thickness (1.870 mm, including the aluminum back). The results show that the SEAL was characterized by an absorption peak with a reduced magnitude of less than 10 dB. This is mainly attributable to the grid of sensors hosted in the spacer, which were made with the lossy coating and consequently partially reflected the EM energy impinging on the panel. Notwithstanding that, the resonance frequency was still centered at 22 GHz. Moreover, the almost unchanged bandwidth (>6.5 GHz at −10 dB) and the −15 dB reflection peak still made the proposed structure a multifunctional screen with significant EM performance.

The other interesting aspect of the prototype is its intrinsic capability for monitoring deformations, as demonstrated in Section 3.2. In fact, the presence of the grid of sensors can be exploited to identify the region where an impact occurs on the structure using the values of resistance variations of the different piezoresistive strips and applying a sensor triangulation-like methodology.

To further support this, we performed several simulations with COMSOL Multiphysics using the 3D FEM model of the sensorized panel, properly stressed at different points. In particular, we will consider the panel clamped at its edges and with its surface divided into nine square areas (A_1_, A_2,_ … A_9_) having centers P_1_, P_2,_ … P_9_, as shown in Figure 20a.

Figure 20b–d show a map of the *xx* component of the conductivity tensor of the piezoresistive sensors when the plate was stressed with a vertical load at points P_1_, P_4_, and P_5_, respectively, such as to determine locally a displacement of 2.5 mm in the opposite direction of *z*. The calculated resistance variations of the sensors are reported in Table 2. Those data show the effectiveness of the system and the potential strategy for identifying the position of the impacted zone. In fact, when the stressed areas were those at the corners, due to a concentrated load in P_1_, P_3_, P_7_, or P_9_, the highest elongation (and consequently the greatest resistance variation) will result for the sensor couples (S_1_, S_2_), (S_2_, S_4_), (S_1_, S_3_), or (S_3_, S_4_), respectively. For impacts occurring on a lateral area (e.g., at P_2_), the significant resistance variation will be of three sensors (i.e., (S_1_, S_2,_ S_4_)). In contrast, when the panel is solicited near the center of the sensor array (i.e., P_5_), the resistance variations of all the sensors will be of the same order of magnitude.

In conclusion, the final device is promising for industrial use in a new category of innovative multifunctional systems.

## 5. Conclusions

This paper focuses on the development of an innovative multi-layered thin panel which integrates nanostructured coatings for distributed sensing and EMI suppression. The coating was produced using a novel GNP-filled polyurethane paint whose piezoresistive and EM properties were tailored through the control of the GNPs concentration and layer thicknesses.

Specifically, a mixture containing 4.5 wt.% of GNPs was optimized for structural monitoring applications, while the electromagnetic-absorbing coating was designed with a filler percentage equal to 8 wt.%

Tests demonstrated the effectiveness of the proposed solution in both distributed SHM and EM-absorption applications. In particular, the produced piezoresistive sensor array showed a high sensitivity for small deformations. Moreover, from numerical simulations it was possible to demonstrate the effectiveness of the sensor array system in impact detection. The SEAL, integrating the sensor array and having an overall thin thickness of 0.870 mm, showed a minimum reflection coefficient of −15 dB at 22 GHz and a bandwidth at −10 dB of 6.5 GHz.

In conclusion, the new SEAL described in this work is an innovative multifunctional structure, able to meet the need, occurring in aviation safety management, for structural health monitoring and electromagnetic compatibility.

Future work could be based on the integration of artificial intelligence (AI) to create data archives of the aviation environment and thus enable the transformation of the maintenance paradigm from condition-based to predictive maintenance [44].

## Figures and Tables

**Figure 1 sensors-22-08470-f001:**
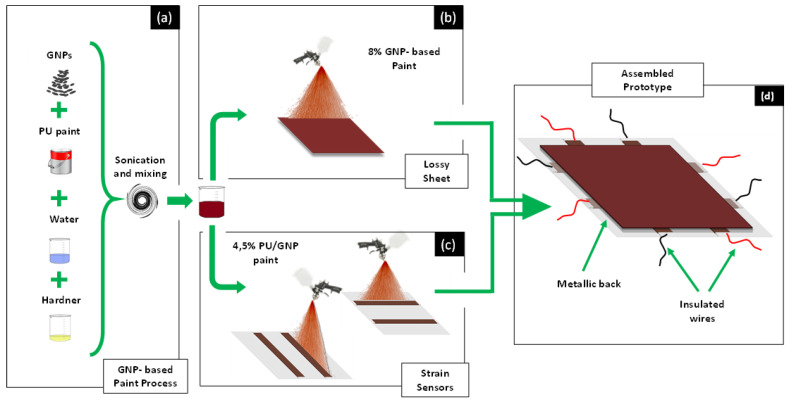
Schematic representation of the fabrication steps of the sensorized radar-absorbing structure: (**a**) production of the paints loaded at different weight concentrations of GNPs; (**b**) realization of the lossy sheet; (**c**) realization of the strain sensor array; (**d**) final assembled structure.

**Figure 2 sensors-22-08470-f002:**
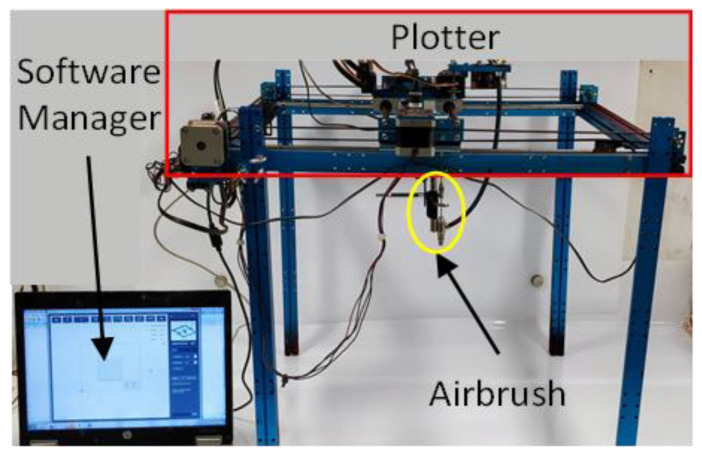
Set-up for the spray deposition of the GNP-based paint.

**Figure 3 sensors-22-08470-f003:**
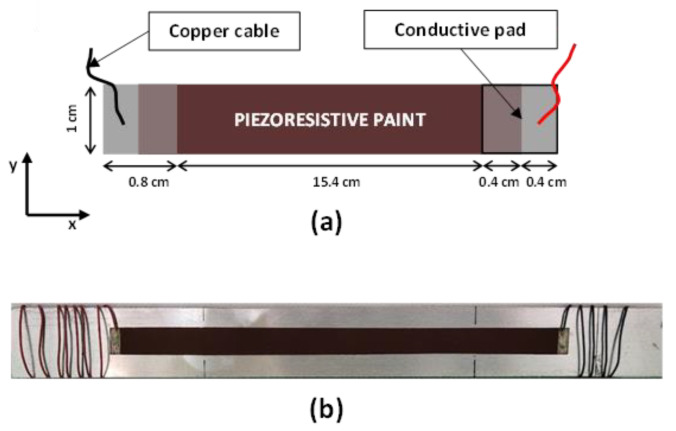
(**a**) Schematic representation of the piezoresistive sensor; (**b**) picture of the sensor realized on a mylar substrate glued onto an aluminum beam for the execution of the electromechanical tests.

**Figure 4 sensors-22-08470-f004:**
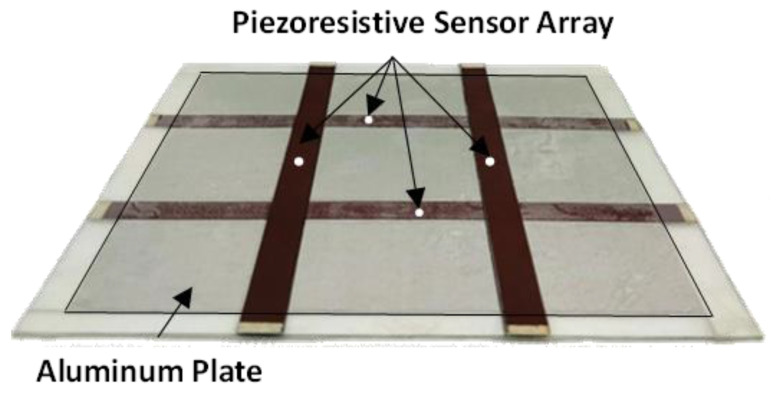
Picture of the grid constituted by the two orthogonal couples of piezoresistive sensors on an aluminum plate.

**Figure 5 sensors-22-08470-f005:**
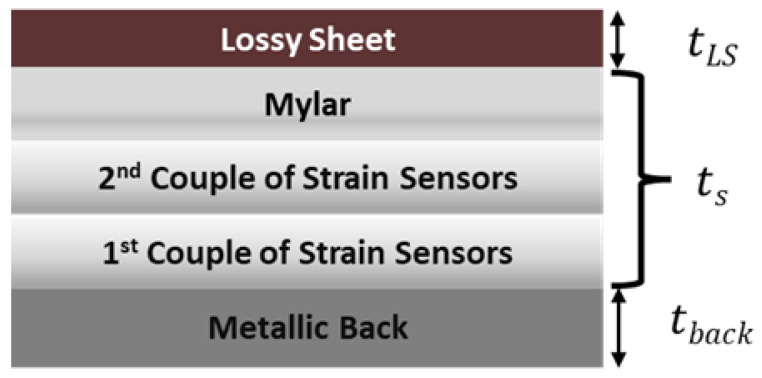
Sketch of the SEAL constituted by the stack of five different layers: the lossy sheet on a mylar substrate (RAS), two mylar layers with sensor strips, and the aluminum back plate.

**Figure 6 sensors-22-08470-f006:**
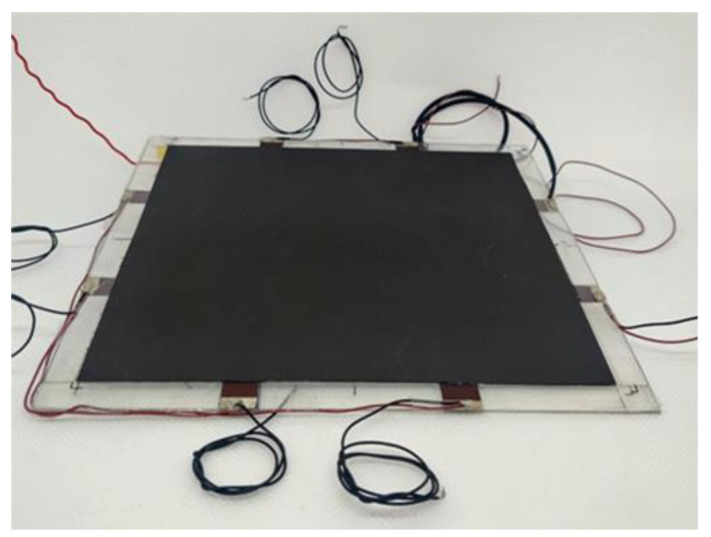
Picture of the final structure (SEAL) after the assembly of the RAS and the layers with the sensor array.

**Figure 7 sensors-22-08470-f007:**
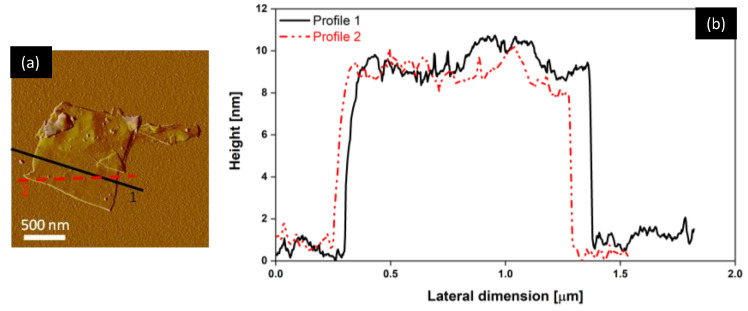
AFM image of the GNPs (**a**) and their corresponding height profiles (**b**).

**Figure 8 sensors-22-08470-f008:**
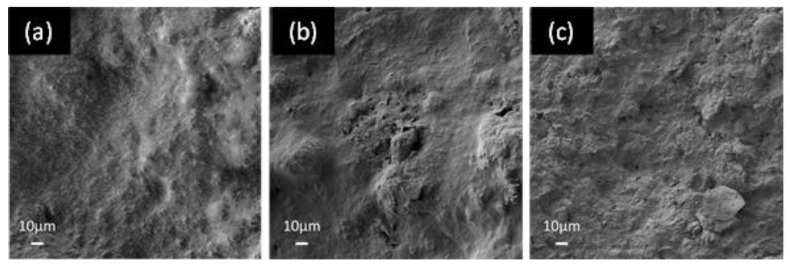
FE-SEM images of the surfaces of: (**a**) the polyurethane unfilled matrix (neat paint); (**b**) the produced sensor with 4.5 wt.% of GNPs; (**c**) the lossy sheet of RAS obtained using 8 wt.% of GNPs.

**Figure 9 sensors-22-08470-f009:**
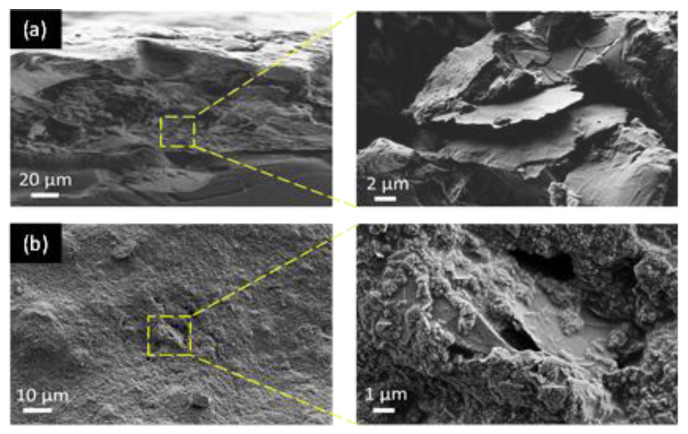
FE-SEM images of the cross section (**a**) and surface (**b**) of the graphene-based paint loaded at 4.5 wt.% of GNPs.

**Figure 10 sensors-22-08470-f010:**
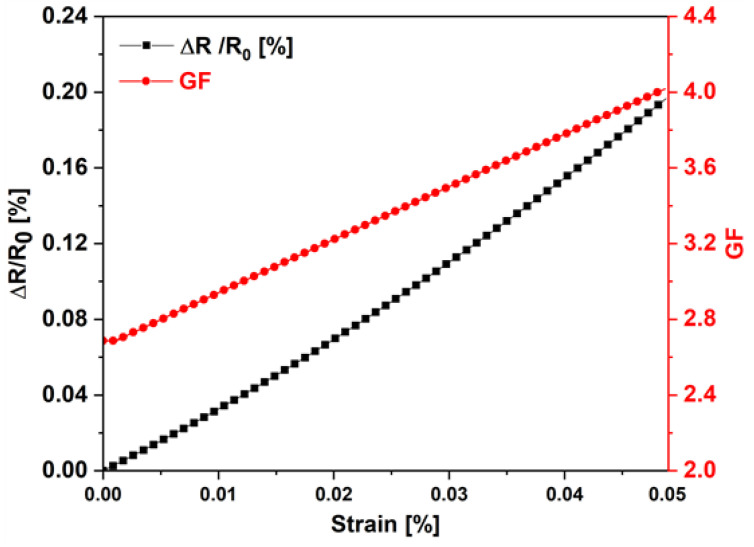
Trends of percentage change of resistance and variation of the gauge factor in the piezoresistive sensor subjected to the three-point bending test, as a function of deformation.

**Figure 11 sensors-22-08470-f011:**
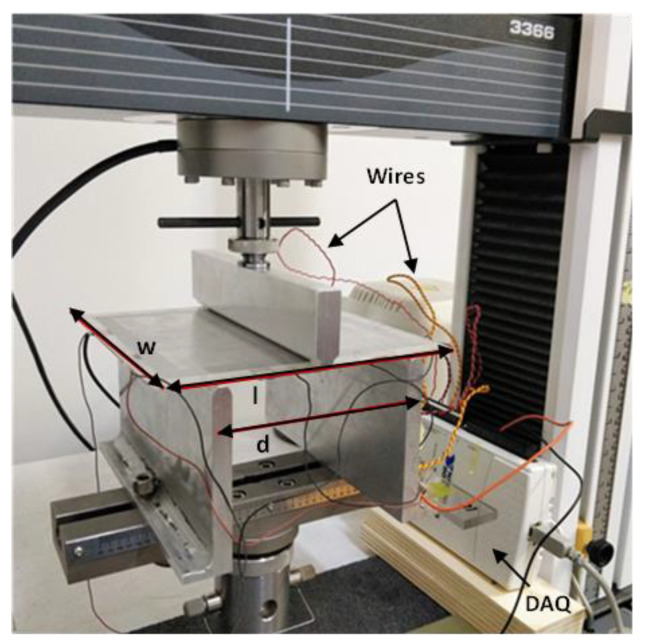
Set-up used to perform the different tests on the prototype with the array of strain sensors (w = 15 cm; l = 15 cm; d = 10 cm).

**Figure 12 sensors-22-08470-f012:**
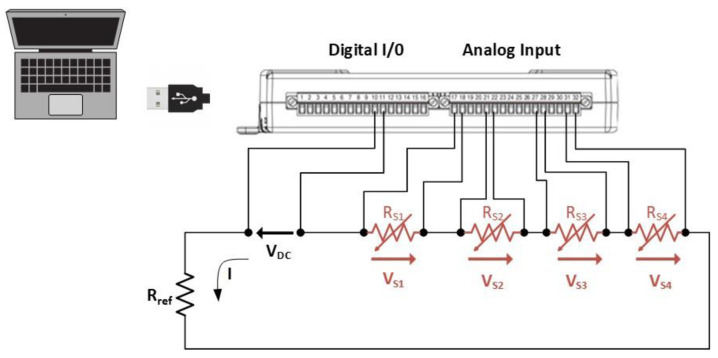
Diagram of connections between sensors and the DAQ. The sensors are represented by the 4 variable resistors $R_{S1},R_{S2},\;R_{S3},\;R_{S4}$; the voltages acquired with the acquisition board are respectively $V_{S1},V_{S2},\;V_{S3},\;V_{S4}$.

**Figure 13 sensors-22-08470-f013:**
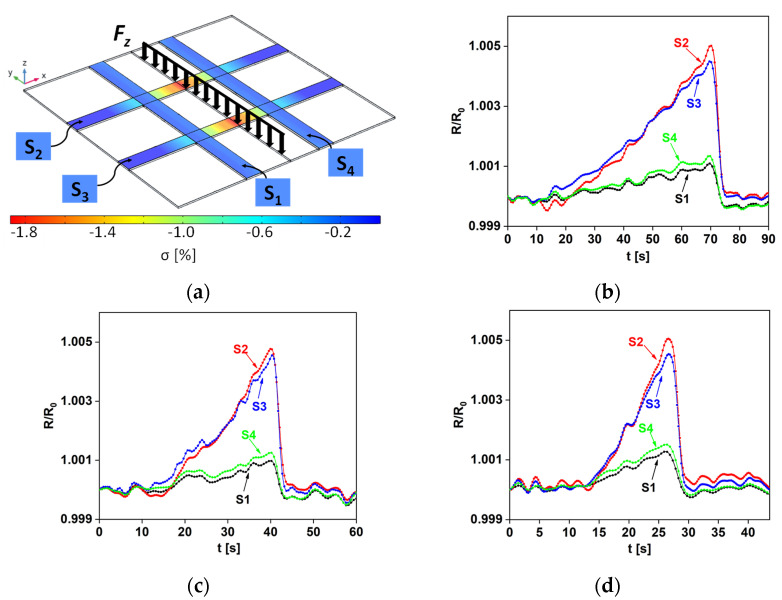
(**a**) Schematic representation of the distributed load applied to the sample with the sensor array and conductivity map of the piezoresistive strips at the maximum induced plate deformation; normalized resistance variation of the four sensors over time at (**b**) 5 mm/min, (**c**) 10 mm/min, and (**d**) 20 mm/min crosshead speed.

**Figure 14 sensors-22-08470-f014:**
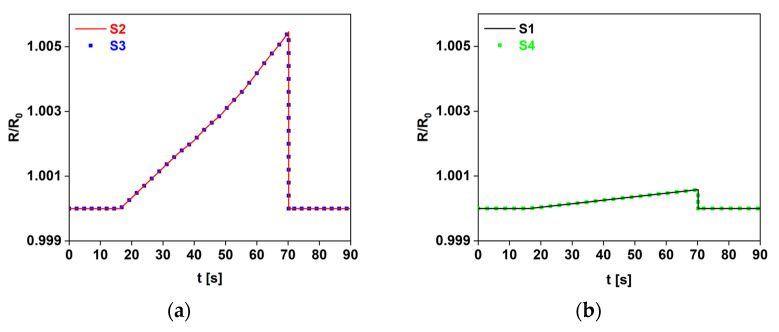
Comparison between the simulation of the sensors perpendicular (**a**) and parallel (**b**) to the load.

**Figure 15 sensors-22-08470-f015:**
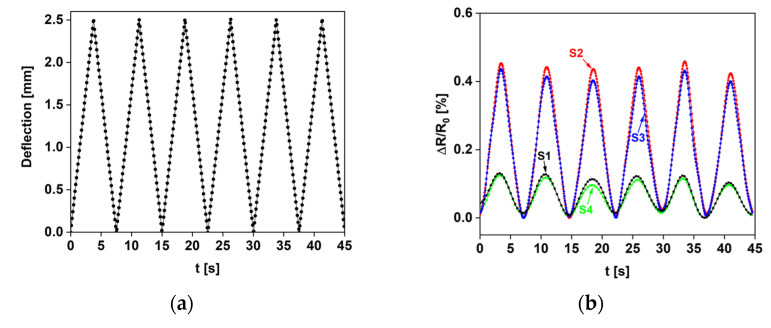
(**a**) Cyclic diagram of displacement impressed by the crosshead. (**b**) Normalized resistance variation of the four sensors over time, with the crosshead moving repetitively from 0 mm to 2.5 mm.

**Figure 16 sensors-22-08470-f016:**
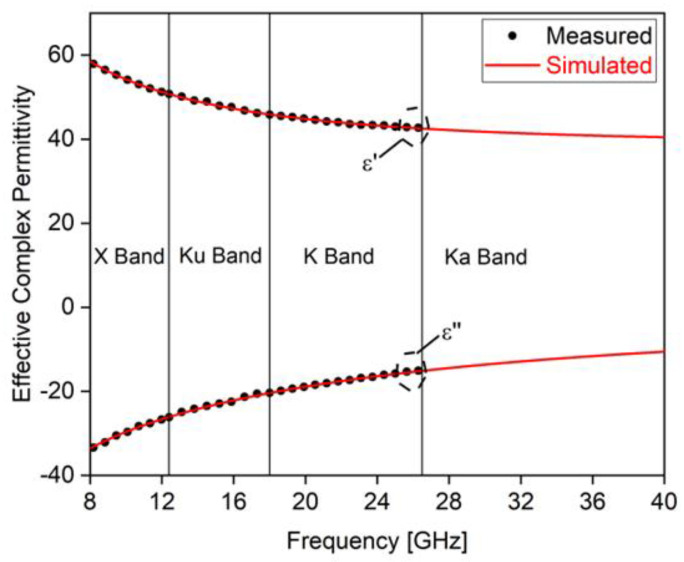
Real and imaginary parts of the complex permittivity of the graphene-based paint from X to K bands and the simulated trend obtained by the Debye’s formula up to 40 GHz.

**Figure 17 sensors-22-08470-f017:**
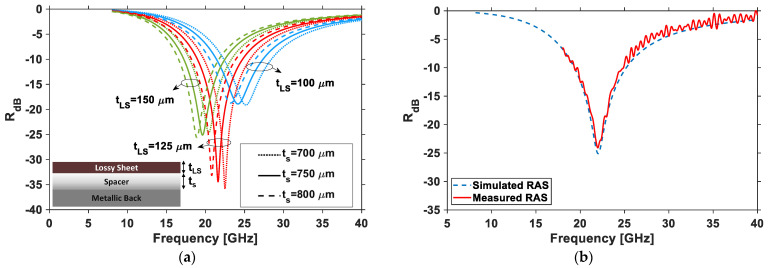
(**a**) Reflection as a function of different combinations of spacer and lossy thicknesses. (**b**) Comparison between measured (red line) and simulated (dashed blue line) reflection coefficients of RAS with *t_s_* = 750 µm and *t_LS_* = 120 µm.

**Figure 18 sensors-22-08470-f018:**
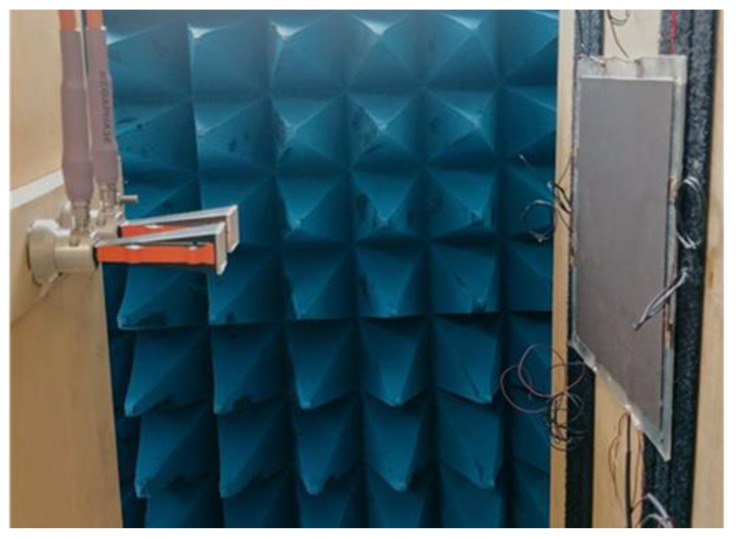
Set-up for the reflection coefficient measurement of the SEAL.

**Figure 19 sensors-22-08470-f019:**
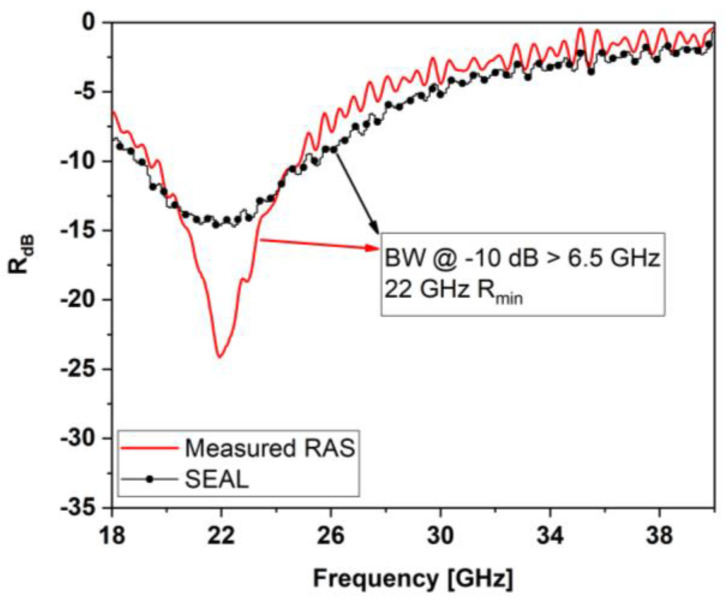
Comparison between the measured reflection coefficient of the SEAL (black line) and that of the RAS without sensors (red line).

**Figure 20 sensors-22-08470-f020:**
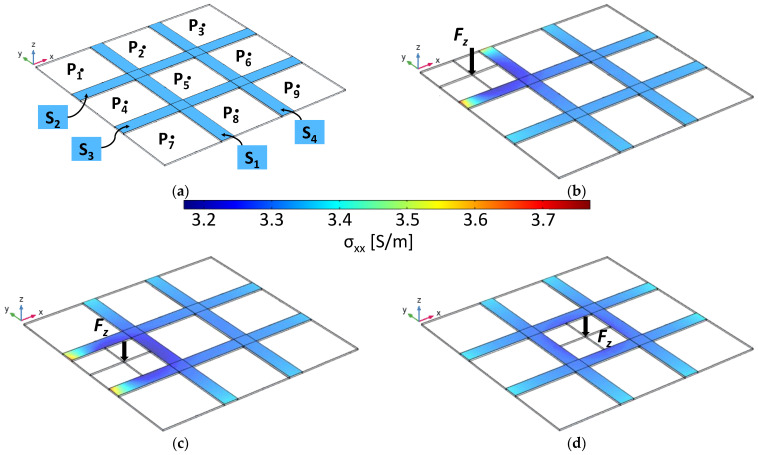
(**a**) Schematic representation of the surface subdivision map. Conductivity maps of the piezoresistive strips at the maximum induced plate deformation of the sensors as a function of the load applied on the zones: P1 (**b**), P4 (**c**), and P5 (**d**).

**Table 1 sensors-22-08470-t001:** Comparison of prior work and current solution.

Paper		Sensing and Electromagnetic Absorbing Properties						
*Ref.*	*Year*	*Functional* *Materials*	*Electromechanical Test*	*Max* *Sensitivity (Gauge Factor)*	*Max Strain*	*Frequency Band* *[GHz]*	*Min.* *Reflection* *Coefficient* *[dB]*	*Overall* *Thickness* *[mm]*
[33]	2021	SiC fiber network embedded composite	Impact	n.a. (8.25 for SiC fibers)	n.a.	8.2–12.4	−38.7 (@9.73 GHz)	3.132
[34]	2022	Ni/CNT decorated Melamine sponge	Tensile	3.06	40%	2–18	−25.7 (average value)	2
[35]	2020	MXene/polymide aerogel	Compression	1.5	50%	3.9–18	−41.8 (@ ~6 GHz)	4
This Work		PU/GNPs paint	Bending	4	0.05%	18–40	−15 (@ 22 GHz)	0.870

**Table 2 sensors-22-08470-t002:** Simulated resistance variation of the four sensors, depending on the force application point.

Point	Δ*R_S_*_1_ (Ω)	Δ*R_S_*_2_ (Ω)
P1	1372.2	1208.0
P4	1082.5	1068.5
P5	749.04	679.65

## Data Availability

Not applicable.

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
