# Peer review of "New Sensing and Radar Absorbing Laminate Combining Structural Damage Detection and Electromagnetic Wave Absorption Properties"

_sensors, 2022, doi:10.3390/s22218470_

Round 1

Reviewer 1 Report

The article is well-written. It is easy to follow the text. The following remarks can be made:

1. There is no characterization of GNPs. Where are the proofs that one has GNPs. TEM, XRD data should be included.

2. It is known, that elongation of the film sensor induces its heating under strain. What is the effect of this heating on sensor response?

3. Section 3.3 needs a comparison with already published data.

4. The advantage of GNP application compared to other materials is not clear.

The article needs major revision.

Reviewer 2 Report

The authors have proposed a method for " New Sensing and Electromagnetic Absorbing Laminate Combining Structural Damage Detection and Radar Absorbing 3 Properties. However, the main contribution of this paper is limited.

1. The keywords are too broad, failing to draw attention to this paper's primary contribution.

2. What is novel about this approach? Such works have been published in great numbers.

3. Authors should compare their approach with other current approaches to the same target for a fair comparison.

4. What is the planned work's motivation? Research gaps and the proposed work's goals should be explicably justified

5. In general, the fundamental backdrop is not presented well, and the notations are not very well illustrated.

4. The study lacks a theoretical framework, which the reader needs in order to understand the study's main points.

7. It is advised that authors expand on their discussion of the findings and provide some rationale for why the chosen strategy is superior to alternative approaches.

8. Has this kind of research ever been attempted? In this essay, support this claim with an appropriate justification.

Reviewer 3 Report

Dear Authors, 

congratulations for the work done.

This paper aims at presenting an innovative system in the field of smart mobility (in particular, aerospace and automotive applications). Developed for aeronautical applications, this system consists of a panel with a multilayer nanostructured coating (i.e., a stack of dielectric and conductive layers including polyurethane paint filled with Graphene Nanoplatelets, GNPs, at different concentrations) that is able to suppressing electromagnetic disturbances (up to 40 GHz) and to monitoring low-energy impacts.

Quasi-static mechanical bending tests and numerical simulations were carried out to test the grid’s strain sensors.

Results show that the proposed system has an EM response that was espressed in terms of minimum reflection coefficient (-20 dB at 22 GHz) and of bandwidth (-10 dB at 6.5 GHz). 

In general, the paper is interesting and well written. 

The following tips and comments have been provided to improve the readability:

1. Abstract: It is better to introduce here the name of the porposed system "Sensing & EM Absorbing Laminate (SEAL)".

2. Please, note that the titles of sections 2 and 3 are the same, i.e. "Materials and Methods". You can change the title of section 3 or consider the content of section 3 as the "Methods" of section 2 (i.e., merge the sections).

3. Equation 1: Please, include the unit of measures of the parameters (even for the remaining equations).

4. Line 262: What did you mean with "xx"?

5. Figures 11 and 12 show peaks with same ordinates, but different abscissa (time)? You should explain why.

Best regards.

Round 2

Reviewer 1 Report

The article can be accepted.